# Does C-C Motif Chemokine Ligand 2 (CCL2) Link Obesity to a Pro-Inflammatory State?

**DOI:** 10.3390/ijms22031500

**Published:** 2021-02-02

**Authors:** Sebastian Dommel, Matthias Blüher

**Affiliations:** 1Medical Department III—Endocrinology, Nephrology, Rheumatology, University of Leipzig Medical Center, Germany Liebigstr. 20, 04103 Leipzig, Germany; sebastian.dommel@medizin.uni-leipzig.de; 2Helmholtz Institute for Metabolic, Obesity and Vascular Research (HI-MAG), Helmholtz Zentrum München, University of Leipzig and University Hospital Leipzig, 04103 Leipzig, Germany

**Keywords:** adipokine, adipose tissue, obesity, inflammation, chemokine

## Abstract

The mechanisms of how obesity contributes to the development of cardio-metabolic diseases are not entirely understood. Obesity is frequently associated with adipose tissue dysfunction, characterized by, e.g., adipocyte hypertrophy, ectopic fat accumulation, immune cell infiltration, and the altered secretion of adipokines. Factors secreted from adipose tissue may induce and/or maintain a local and systemic low-grade activation of the innate immune system. Attraction of macrophages into adipose tissue and altered crosstalk between macrophages, adipocytes, and other cells of adipose tissue are symptoms of metabolic inflammation. Among several secreted factors attracting immune cells to adipose tissue, chemotactic C-C motif chemokine ligand 2 (CCL2) (also described as monocyte chemoattractant protein-1 (MCP-1)) has been shown to play a crucial role in adipose tissue macrophage infiltration. In this review, we aimed to summarize and discuss the current knowledge on CCL2 with a focus on its role in linking obesity to cardio-metabolic diseases.

## 1. Introduction

Accumulation of adipose tissue (AT) is the major symptom of obesity. Until about 25 years ago, AT was regarded as an energy storage organ that additionally acts as isolation for the inner organs [1]. Due to the discovery of its endocrine function in the late 1980s, our understanding of AT changed fundamentally [2]. Since then, hundreds of bioactive components secreted by AT have been found [3,4]. Those AT-derived secretory factors including leptin, adiponectin, adipsin, plasminogen activator inhibitor-1 (PAI1), complement components, or cytokines such as tumor necrosis factors (e.g., TNF-α) or chemokines (e.g., CCL2) have been described with the term “adipokines” [5]. 

In 1999, Funahashi et al. defined “*biologically active molecules secreted from adipose tissue*” as “adipocytokines” [6]. However, this term is potentially misleading because it suggests that all AT-secreted substances are cytokines. While this is true for some AT-secreted molecules (e.g., IL-6 or TNF-α), the majority is of non-cytokine origin. Although Trayhurn and Wood recommended to use the term “*’adipokine’ […] to describe a protein that is secreted from […] adipocytes*”, commonly all AT-produced and -secreted substances are named “adipokines”, independent of whether they are secreted primarily from adipocytes or other AT cell types [7]. 

Adipokines are a heterogenous group of peptides including hormones, growth factors, and cytokines which differ in their physiological functions. Adipokines play an important role in the regulation of energy homeostasis, appetite, satiety, lipid metabolism and glucose homeostasis, blood pressure and vascular homeostasis, angiogenesis, and immune response [8]. Whereas adipocyte-secreted adiponectin and leptin circulate in the blood as endocrine factors, it was demonstrated that some adipokines mainly have a para or autocrine functions within AT without a contribution to inter-organ tissue crosstalk [9]. Serum concentrations of several adipokines reflect body energy stores, fat mass and distribution, systemic insulin sensitivity, glucose tolerance, a pro-inflammatory state, and other phenotype characteristics [4,10,11,12,13,14,15,16]. As examples, leptin serum concentrations are proportionally secreted to body fat mass [17], where circulating adiponectin are typically lower in individuals with obesity compared to those who are lean [18]. Additionally, several immune-modulating adipokines, such as IL-6, IL-8, CXCL5, or CCL2, are elevated in the obese state [9,19]. These changes in adipokines’ secretion pattern can be explained by AT remodeling, a process in which quantitative and qualitive changes in the cellular composition of AT occur in response to excessive weight gain [20].

AT is a complex organ composed of several cell types (Figure 1). Adipocytes account for up to 80% of AT volume, but reflect only 20–40% of cell number. AT consists of adipose-derived stem cells (ADSCs), preadipocytes, endothelial cells, and leukocytes [21,22]. Very recently, single-nucleus RNA-sequencing (snRNA-seq) analysis of mouse and human adipose tissue revealed a subpopulation of adipocytes that regulates thermogenesis [23]. Depending on their type, different AT-cells produce distinct adipokine patterns. Therefore, knowing the cellular origin for adipokine production is important to dissect which cell type might be enriched and/or activated in AT. For example, adipocytes exhibit a quantitatively distinct adipokine pattern in the function of the fat-depot (subcutaneous (sc) and visceral (vis)) origin. Adiponectin and leptin are predominantly expressed in sc AT [24]. In contrast, IL-6 [25], omentin [26], visfatin [27], and RBP4 exhibit higher vis than sc production [28]. Adipsin [29], lipocalin 2 [30], and TNF-α [31] are secreted in both depots in comparable amounts. Using single-cell or single nuclei transcriptomics, it is now possible to discriminate adipocyte subpopulations within one depot, as well as more than 10 different AT cell types which differ in their metabolic and transcriptional properties including the identification of differences in cellular adipokine sources [23,32,33,34].

Besides adipocytes, ADSCs produce a variety of chemokines and growth factors such as the pro-angiogenic CCL2, IL-8, vascular-endothelial growth factor (VEGF) [35], platelet-derived growth factors (PDGF) [36], or c-kit which induces endothelial cell proliferation [37]. Furthermore, ADSCs secrete immune-modulating substances like interferon-γ (IFN-γ) or transforming growth factor-β (TGFβ) [38].

Depending on the fat depot, 15–50% of all resident AT-cells are preadipocytes [39]. Using a conditioned medium from obese murine epididymal AT, Renovato–Martins et al. demonstrated that 3T3-L1 preadipocytes secrete leptin and adiponectin as well as the pro-inflammatory factors IL-6, TNFα, and IL-1β [40].

In addition to those cell types of mesenchymal origin, there are various hematopoietic cells resident in AT. Nearly all known leukocytes such as macrophages, monocytes, dendritic, or natural killer cells, B-, and T-cells, as well as neutrophils or eosinophils, are of high importance in the adipokine context. The majority of immune cells express the leptin receptor on their cellular surface. Since circulating levels elevate proportional to the amount of white adipose tissue, leptin acts as a pro-inflammatory adipokine on immune cells. Subsequently, leptin receptor signaling via JAK2-STAT leads to a broad range production of pro-inflammatory adipokines, such as interleukins (IL-6, IL-8, IL-12, and IL-18), TNFα, or CCL2 [22,41]. Indeed, CCL2 (MCP-1) is a member of the small inducible gene family that plays a role in the recruitment of monocytes to sites of injury and infection, but also to AT under conditions of inflammation or adipocyte apoptosis [42,43,44]. Recently, CCL2 has been described as an important factor linking sc AT to altered glucose metabolism and body fat distribution [45].

In this review, we summarized recent data on the importance of C-C motif chemokine ligand 2 (CCL2) in the context of obesity.

## 2. C-C Motif Chemokine Ligand 2

### 2.1. Structure, Sources and Signaling

Chemokines are proteins with molecular weights ranging between 8 to 12 kDa that mediate cellular movement (chemotaxis), hematopoiesis, leukocyte degranulation, and angiogenesis [46]. Four chemokine subfamilies have been categorized based on the number and location of N-terminal cysteine residues: C, CC, CXC, and CX3C [47]. Chemokine sequences are highly conserved and share similar structures consisting of a flexible N-terminus followed by a loop containing three antiparallel β-sheets on to which is folded an α-helix [48]. Experiments which also defined the crystal structure of CCL2 revealed that it forms dimers in an anti-parallel β strand arrangement between the two flexible N-termini [49]. 

CCL2, also known as monocyte chemoattractant protein-1 (MCP-1), was the first discovered human CC-family chemokine [50,51]. The gene is located on chromosome 17 (q11.2) [52] and is produced by endothelial cells, fibroblasts, epithelial, smooth muscle, mesangial, astrocytic, monocytic, and microglial cells [53,54,55,56], whereas monocytes and macrophages are major sources of CCL2 [57,58]. Additionally, (pre-)adipocytes express CCL2 [59]. 

CCL2 expression is induced by inflammatory stimuli (IL-1, IL-4, IL-6, tumor necrosis factor α (TNFα)), transforming growth factor β (TGFβ), lipopolysaccharide (LPS), interferon γ (IFNγ), platelet-derived growth factor (PDGF), vascular endothelial growth factor (VEGF), macrophage colony-stimulating factor (M-CSF), and granulocyte-macrophage colony-stimulating factor (GM-CSF) [60]. Human serum CCL2 has been associated with a chronic pro-inflammatory state and was suggested as a biomarker for malignant disease such as prostate and breast cancer [61,62]. High *CCL2* expression in tissues indicates chemo-attraction of monocytes in the context of local defense mechanism activation and repair of tissue damage [63]. 

Chemokines are secreted in response to pro-inflammatory signals, such as cytokines, to selectively recruit immune cells including monocytes, neutrophils, or lymphocytes to sites of inflammation or injuries. For CCL2, there are two activation pathways described. During the canonical pathway, inflammatory substances such as tumor-necrosis factor α (TNFα) binds to its receptor, resulting in the activation of the inhibitor of nuclear factor-κB kinase (IKK). Activated IKK then phosphorylates the NF-κB-bound inhibitor of NF-κB (IκB), whereby IκB is degraded. Consequently, released NF-κB homodimers translocate to the nucleus where they activate the transcription of inflammation-related genes e.g., *CCL2*, *TNFα*, and *IL-6* [64]. Alternatively, CCL2 can be activated by the non-canonical pathway, i.e., NF-κB-independent CCL2 expression stimulated by PDGF [50] or insulin. In human aortic endothelial cells, physiological insulin concentrations were shown to suppress the expression of both *NF-κB* and *CCL2* by more than 60% [65]. Nakatsumi et al. demonstrated that insulin activates the phosphatidylinositol 3-kinase (PI3K)-Akt pathway which leads to the inhibition mTORC1-repressor, the ras homolog enriched in brain (RHEB). In turn, mTORC1 induces forkhead box K1 (FOXK1) dephosphorylation via protein phosphatase 2A (PP2A), leading to *CCL2* expression [66] (Figure 2).

The effects of CCL2 on target cells are mediated by a specific chemokine receptor. Cells that express the distinct CC chemokine receptor (CCR) are able to migrate along the chemokine gradient upon CCL2 activation [67]. CCRs are G-protein coupled receptors (GPCRs) belonging to the rhodopsin or serpentine receptor family [68] which are expressed on different types of leukocytes such as eosinophils, basophils, lymphocytes, macrophages, and dendritic cells [69]. Human CC chemokines bind to at least two different CCRs.

CCL2 usually binds to CCR2 that exists in two different splice variants, CCR2A and CCR2B, which differ in their C-terminal tails [70]. In contrast to the widespread expression of CCL2, CCR2A is mainly expressed by vascular smooth muscle cells and mononuclear cells, whereas CCR2B is the predominant receptor on monocytes and natural killer cells [71]. In addition to CCL2, CCR2 binds another five pro-inflammatory cytokines, CCL7 [72], CCL8 [73], CCL12 [74], CCL13 [75], and CCL16 [76]. However, CCL2 has the highest activation potential that finally leads to monocyte migration into target tissues [77]. As a result of CCL2/CCR2 binding, cell migration is promoted by the activation of several signaling cascades such as JAK2/STAT3 [78], MAP kinase [79], and PI3K [80] pathways (Figure 3).

CCL2’s binding at CCR2 results in the dissociation of GDP from the G_αi_ subunit which associates with intracellular GTP and inhibits membrane-bound adenylyl cyclases, finally leading to decreased intracellular cAMP levels. In contrast, the released G-protein βγ heterodimer activates phospholipase C which then hydrolyzes phosphatidylinositol 4,5-bisphosphate (PIP_2_) to diacylglycerol (DAG) and inositol 1,4,5-trisphosphate (IP_3_) [75]. IP_3_ diffuses in the cytosol and stimulates calcium release from the endoplasmic reticulum [81]. The released Ca^2+^ is further bound by calmodulin (CaM), an essential modulator of various processes like immune response, inflammation [82], apoptosis, or metabolism [83]. Elevated Ca^2+^ levels as well as DAG activate protein kinase C-β (PKC-β) that mediates gene expression via c-Jun N-terminal kinases (JNK) and extracellular signal-regulated kinases (ERK) activation [84]. Monocyte migration is regulated via G_βγ_ activation of PI3K/Akt, which in turn polymerizes actin for pseudopod formation [85].

The multiple and pleiotropic effects of CCL2 on multiple cells of the myeloid lineage are summarized in Figure 4 and are more extensively discussed in a recent comprehensive review highlighting the role of CCL2 on immune cell behavior and tumor immunity [60]. In our review, we focused on the role of CCL2 in the context of obesity-related diseases.

### 2.2. Animal Models

To study the physiological role of CCL2 and to investigate its signaling pathways, knockout mice were generated. Mice deficient in *Ccl2* are viable, fertile, and reproduce with a normal litter size, sex distribution, development, and life span as expected from wild-type mice [42,87].

Mice with an ablation of *Ccl2* develop severe defects in monocyte recruitment to sites of inflammatory damage or in response to immunological signaling [42]. When crossed with low density lipoprotein (LDL) receptor-deficient mice, CCL2 has been shown to reduce atherosclerosis upon high-cholesterol diets, suggesting an important role of CCL2 in the initiation of atherosclerosis [88].

Even though CCL2 has a central role in monocyte recruitment, there are several CC chemokines known that modulate chemotaxis in a similar manner, e.g., CCL7 (MCP3), CCL8 (MCP2), CCL12 (MCP5), or CCL14 (MCP4). Notwithstanding that Ccl2 mainly binds to Ccr2, this receptor can be further activated by Ccl7 and Ccl12 in mice [89,90,91]. In summary, this suggests that other chemokines potentially compensate for the loss of *Ccl2* in knockout mice. Nevertheless, *Ccl2*^−/−^ mice showed impaired monocyte trafficking and cytokine secretion. Apart from finding normal numbers of Kupffer cells and macrophages in *Ccl2*-deficient mice, these mice were not able to recruit monocytes or macrophages 72 h after thioglycolate administration [42], an agent that increases monocyte migration after intraperitoneal injection [92]. Splenocytes of *Ccl2*^−/−^ mice were characterized by a significant reduction of IL-4 and IL-5 production and an about 50% reduced IFN-γ secretion [42].

In a model of myocardial infarction, *Ccl2*^−/−^ mice were characterized by both a decreased and delayed macrophage infiltration and a delayed replacement of damaged cardiomyocytes [93]. Furthermore, these mice showed similar infarct sizes, an extended inflammation phase, and a lower post-infarction left ventricle remodeling [93].

Knockout mice were further used to validate the impact of Ccl2 in ulcerative colitis. *Ccl2*^−/−^ mice exhibit lower macrophage and CD3^+^ T cell infiltration as well as reduced IL-1β, IL-12p40, and IFN-γ production, ultimately resulting in a reduced severity of colitis [94].

Targeting Ccl2 as a therapeutic strategy in kidney diseases was also analyzed using mice lacking *Ccl2*. Nephritis was induced with nephrotoxic serum, whereupon wild-type mice reacted with five times higher *Ccl2* expression compared to unstimulated mice [95]. Around 90% of renal Ccl2 was localized within cortical tubules and most of them get damaged during induced nephritis. In contrast, *Ccl2*-deficient mice presented a ~40% reduction of tubular injury coming from a decreased macrophage recruitment to Ccl2 producing tubular endothelial cells [95].

Furthermore, Ccl2 plays an essential role in skeletal muscle regeneration. The comparison of wild-type and *Ccl2*^−/−^ mice which underwent femoral artery excision (FAE) exposed a decreased, but longer lasting macrophage infiltration associated with residual necrotic tissue. The injured muscle was mostly regenerated 21 days after FAE in *Ccl2*^−/−^ mice, but in contrast to wild-types they presented an enhanced adipocyte accumulation within the remodeled muscle tissue [96]. Both decreased macrophage infiltration and impaired muscle regeneration with increased adipocyte infiltration were also described for Ccr2-deficient mice [97].

### 2.3. Human Mutations in the CCL2 Pathway and Associated Diseases

Polymorphisms within the *CCL2* gene are described in the context of several human diseases. Neural tube defects (NTD) such as spina bifida belong to the most common congenital malformations that can be well treated by maternal folic acid supplementation during pregnancy [98]. Nevertheless, a maternal polymorphism in the *CCL2* promotor region is associated with a 1.5-fold increase of NTD in children [98]. Additionally, a chronic low folate status is a major factor to suffer from hyperhomocysteinemia which, in turn, is a risk marker for atherothrombotic diseases. During the development of atherosclerotic lesions, endothelial cells upregulate the expression of *CCL2*. Therefore, folate shortage results in increased CCL2 levels via augmented *p38* expression [99]. Moreover, women carrying a 677C>T polymorphism in methylenetetrahydrofolate reductase, a key enzyme in the folate/homocysteine metabolism, showed strongly elevated CCL2 levels caused by enhanced homocysteine levels [100,101].

Two single nucleotide polymorphisms (SNP) located in the *CCL2* promotor region (-2136T) and in intron 1 (767G) are found strongly associated with an reduced susceptibility to human immunodeficiency virus 1 (HIV-1) infection [102]. Because CCL2 is not able to bind the HIV-1 coreceptors CCR5 and CXCR4, the authors suggested that carriers of those SNPs potentially exhibit an differential immune response to HIV-1 infection [102].

Several SNPs in the *CCL2* gene are associated with autoimmune diseases. As an example, the -2518A/G variation within the *CCL2* promotor region has been shown to be associated with Crohn’s disease without affecting CCL2 plasma levels [103]. In contrast, ulcerative colitis presents raised CCL2 levels. Here, the -2518A/G SNP alone does not represent a genetic risk factor. However, in combination with a polymorphism within the interleukin-1β gene, the ulcerative colitis risk is increased significantly [104].

Polymorphisms concerning CCL2 are further associated with premature coronary artery disease [105], rheumatoid arthritis [106], sepsis [107], and lupus nephritis [108].

CCL2 has been shown to play an important role during development and progression of diabetic nephropathy (DN). In patients suffering from DN, microRNA miR-374a was found to be downregulated in nephropathic tissue. In cell culture, miR-374a can downregulate *CCL2* expression, which suggests the miR-374a/CCL2 axis as potential target for DN treatment or therapy [109].

### 2.4. Effects of Antibody Administration Affecting CCL2/CCR2 Signaling

CCL2 regulates the monocyte and macrophage migration and infiltration into inflamed, but also tumor tissues. Therefore, a variety of cancers, e.g., glioma [110], breast tumors [111], or prostate cancer [112] are associated with increased serum concentrations of CCL2. Physiological anti-tumor responses can be inhibited by tumor-associated macrophages (TAMs) or myeloid-derived suppressor cells (MDSCs), which promote tumor growth [110]. Systemic CCL2 blockade with anti-CCL2 antibodies (Ab) resulted in decreased TAMs and MDSCs as well as modestly prolonged survival both in mice bearing intracranial GL261 glioma or intracranial human U87 glioma xenografts. However, a combined treatment with the chemotherapeutic agent temozolomide and antibodies resulted in a significantly prolonged survival [110]. A combined treatment was also found to be beneficial for prostate cancer regression. The combination of anti-CCL2 Ab with docetaxel was shown to generate a more effective tumor regression than either Ab- or docetaxel-treatment alone [112]. Nevertheless, the use of CCL2 blockade as therapeutic is discussed controversially. In nude mice bearing MCF10CA1d breast tumor xenografts, continuous delivery of human CCL2-neutralizing Ab (0.3 mg/kg/day using osmotic pumps) was analyzed over 4 weeks. There, tumor growth, macrophage recruitment, and tumor angiogenesis were not affected by CCL2 blockade. Surprisingly, human CCL2 levels were significantly increased in both circulating blood and tumor interstitial fluid, whereas murine CCL2 levels were not affected [111].

Chronic inflammation is accompanied by elevated CCL2 levels as exemplified in hepatocellular carcinoma or rheumatoid arthritis. Mice carrying a miR-122 knockout displayed upregulated hepatic *CCL2* expression leading to hepatitis and hepatocellular carcinoma. Treatment with neutralizing CCL2 Ab suppressed chronic liver inflammation, reduced liver damage and both liver carcinoma incidence and tumor burden by downregulating pro-inflammatory pSTAT3, c-MYC, and NF-κB signals. Tumorigenesis was inhibited by enhanced natural killer cell cytotoxicity and IFNγ secretion after CCL2 Ab administration [113]. However, in the context of rheumatoid arthritis, treatment with human anti-CCL2 monoclonal Ab does not have a benefit compared with placebo control [114]. There was an unexpected dose-related CCL2 increase, resulting in worsening rheumatoid arthritis in patients treated with high doses of the Ab [114]. In addition, the blockade of CCL2 receptor CCR2 via human CCR2 blocking antibodies displayed up to 94% reduction of free CCR2 on monocytes, but without an amelioration of synovial inflammation in rheumatoid arthritis [115]. Taken together, these results did not support a beneficial role of CCL2 Ab treatment in the context of rheumatoid arthritis.

### 2.5. CCL2 in Obesity and Obesity Related Diseases

Obesity is a major risk factor to develop noncommunicable diseases (NCDs) such as hypertension, cardiovascular diseases, type 2 diabetes, and specific types of cancer. Worldwide, all NCDs together account for more than 70% of premature deaths, which in turn highlights the impact of obesity in the context of a global health burden (reviewed in [116]). In general, the BMI correlates with AT *CCL2* expression, whereas weight loss reduces these levels [117]. CCL2 has been shown to play a unique role among several cytokines that may influence the function of adipocytes, recruitment of AT macrophages, and the link between AT inflammation and insulin resistance [117,118,119].

#### 2.5.1. CCL2 Reflects a Pro-Inflammatory State

Obesity can contribute to local AT inflammation which most likely underlies a systemic chronic low-grade inflammation [120]. The typical symptoms of inflammation, heat, pain, redness, and swelling are caused by the effects of inflammatory regulators and mediators such as cytokines or chemokines [121].

The mechanisms of how obesity contribute to activation or maintenance of inflammation are not completely understood, but may include hypoxia in adipose tissue, increased adipocyte apoptosis, several stresses in AT, and others [120,122,123,124]. A permanent excess of nutrients can induce intracellular stress in adipocytes, leading to inflammation [125]. In addition, a systemic pro-inflammatory state is frequently induced by macrophage recruitment as a consequence of adipocyte apoptosis, AT-derived bacteria, accumulation of xenobiotics, altered fatty acid flux within AT, or others [126,127,128,129]. Cell culture experiments revealed the ability of macrophages to express higher levels of *CCL2* mRNA after incubation with palmitic acid [130]. Elevated levels of free fatty acids can upregulate *CCL2* expression through the toll-like receptor (TLR) 4/TIR-domain-containing adapter-inducing interferon-β (TRIF)/interferon regulatory factor (IRF) 3 pathway [131]. Human monocytic cells undergoing a combined incubation with palmitate and TNF-α expressed significantly higher amounts of *CCL2* as treated with one of these components alone [131]. The authors concluded that palmitate binding at TLR4 amplifies the TNF-α/NF-κB-induced *CCL2* expression via downstream TRIF/IRF3 activation.

Moreover, in vitro experiments identified ten miRNAs dysregulated during obesity which are strongly associated with the secretion of CCL2 [132]. miR-193b was shown to modulate *CCL2* expression indirectly by the downregulation of several transcription factors, e.g., *NFKB1*. In contrast, miR-126 can directly bind at the 3′-untranslated region of CCL2 mRNA and thereby regulate its expression [132].

Additionally, both adipocyte hyperplasia and hypertrophy cause local hypoxia within AT that triggers adipocyte death, inflammation, tissue fibrosis, and angiogenesis [133]. In murine 3T3-L1 cells as well as in human SW872 adipocytes cultured under intermittent hypoxia conditions, mRNA levels of *CCL2*, *RETN*, and *TNFα* were significantly increased without affecting promotor activity. All three genes exhibited a miR-452 target sequence and miR-452 levels were significantly decreased under hypoxia in these cells [134]. Hence, the authors summarized that hypoxia downregulates miR-452 which in turn increases the expression of *CCL2*, *RETN*, and *TNFα*, causing insulin resistance. In a comparison of age-, sex-, and BMI-matched patients with metabolically healthy obesity either with preserved insulin sensitivity or insulin resistance, we found that neither CCL2 serum concentrations nor AT expression are related to AT inflammation [14]. Despite that, CCL2 could represent one of the mechanistic links between obesity and related diseases which are at least in part mediated by a pro-inflammatory state.

Very recently, upregulation of CCL2 expression in human subcutaneous AT has been related to AT senescence in severely obese individuals [45]. Together with other factors, including senescence-associated β-galactosidase (SA-β-gal), insulin-like growth factor binding protein 3 (IGFBP3), plasminogen activator inhibitor-1 (PAI-1), and IL-6, CCL2 could thereby contribute to a higher number of senescent cells which may perpetuate AT inflammation and fibrosis as additional cellular sources of proinflammatory factors [45]. An intact remodeling of the extracellular matrix (ECM), a network of different proteins and proteoglycans, is required for healthy adipose tissue expansion [135,136]. Through limiting the expansion of “metabolically safe” fat depots, altered extracellular matrix composition may indirectly contribute to metabolic diseases [137]. AT fibrosis maybe considered an end stage of ECM alterations, which has been associated with obesity comorbid disorders [138,139].

CCL2 binds to proteoglycans, which are part of the extracellular matrix surrounding adipocytes. Whereas lean AT expresses low proteoglycan levels, expression increases with obesity. Proteoglycans immobilize CCL2 and present it to macrophages resulting in higher AT inflammation [140,141,142] (Figure 3). Inhibition of NLRP3 and subsequent reduction of AT *CCL2* expression has been shown to reduce fibrosis and related AT inflammation [143]. In addition to monocyte recruitment into AT, the CCL2-CCR2 axis has been identified as an important mechanism of how AT inflammation in visceral depots may recruit regulatory T cells in a sex hormone-dependent manner [144].

Stromal CCL2 signaling in AT may even play a role in promoting fibrosis of mammary tumors through the recruitment of myeloid-lineage cells [145]. This example demonstrates that increased CCL2 may link obesity, AT inflammation, and fibrosis to certain types of cancers.

#### 2.5.2. CCL2 and Insulin Resistance

It is well established that chronic low-grade activation of the innate immune system caused by obesity leads to the manifestation of insulin resistance and type 2 diabetes. Developing insulin resistance is closely related to adipocyte hypertrophy and following proinflammatory responses [146]. Using a 3T3-L1 cell model, it has been demonstrated that adipocytes treated with saturated fatty acids produced higher levels of proinflammatory cytokines, such as Ccl2, Tnf-α, or Il-6, than cells treated with monounsaturated fatty acids [146]. In contrast, Kim et al. showed that hypertrophic adipocytes became insulin resistant independently of proinflammatory responses through impaired Glut4 trafficking [146]. Furthermore, Kawano et al. found that the first organ expressing high levels of *Ccl2* in response to a high-fat diet (HFD) is the colon. Higher Ccl2 levels recruit pro-inflammatory macrophages, resulting in increased gut permeability, activation of the inflammasome, and finally in inflammation and AT insulin resistance [147].

Rodent studies suggested that insulin resistance is caused by AT inflammation [44,148]. Indeed, immuno-compromised mice are not protected against HFD-induced insulin resistance. In particular, early onset of HFD-induced insulin resistance was independent of inflammation, whereas the chronic state during obesity was mediated by macrophages [149]. However, *Ccr2* knockout mice showed improved insulin resistance and reduced macrophage infiltration under HFD [150].

During obesity, the number of pro-inflammatory M1 macrophages is increased in white adipose tissue (WAT) [151]. This increase can be explained by at least two mechanisms. First, mature adipocytes secrete CCL2 which recruits circulating monocytes into AT where they differentiate into macrophages [151]. The second explanation is that CCL2 triggers resident AT macrophage proliferation during obesity [152]. To investigate the mechanisms by which *Ccl2* is activated in adipocytes, insulin resistant, mTorc2-deficient AdRiKO mice have been generated. Ablation of insulin/mTorc2 signaling resulted in elevated *Ccl2* expression exclusively in adipocytes, but not in fibroblasts or hepatocytes leading to AT inflammation. Based on these findings, the authors proposed that insulin resistance is the cause and not the result of AT inflammation [153].

#### 2.5.3. CCL2 and Cardiovascular Diseases

Obesity is one of the major risk factors leading to the development of cardiovascular diseases. However, the underlying mechanisms are not fully understood yet. Chemokines are of relevance in the pathogenesis of atherosclerotic cardiovascular diseases (ASCVD) and act in a network instead of a single cytokine modality. CCL2 is able to recruit different cell types such as monocytes, memory T cells, or dendritic cells and is therefore associated with cardiac diseases like ischemia, reperfusion injury, or fibrosis heart failure [154]. Plasma levels of CCL2 were found to be significantly increased in patients with coronary heart disease compared to healthy individuals [155].

In a cohort of 2270 patients with acute coronary syndromes, increased CCL2 levels were found to be associated with an increased risk of atherosclerosis and mortality of myocardial infarction [156]. In a study including 1411 patients with ASCVD over a median follow-up of 3.3 years, both lower and higher serum CCL2 levels were shown to be linked to a higher mortality [157]. Another study investigated the influence of the CCL2-2518 A/G SNP of circulating C-reactive protein (CRP) levels in patients with ASCVD. The highest CRP levels were found in patients homozygous for this GG polymorphism [158]. This SNP was further shown to be associated with higher CCL2 levels in patients with insulin resistance compared to those with insulin sensitivity [159]. However, studies investigating circulating CCL2 levels as predictors for ASCVD are not coherent. In a small group of 83 patients with acute myocardial infarction and 38 patients with stabile angina, both CCL2 and CC chemokine, regulated upon activation, normal T cell-expressed and presumably secreted (RANTES) serum levels were analyzed. Whereas CCL2 concentrations were observed as highly variable, the authors suggest RANTES serum levels as better reflectors for atherosclerotic lesions [160]. In summary, it remains controversial whether CCL2 plays a causative role in ASCVD.

### 2.6. CCL2 as Drug Target

CCL2 is a promising drug target for inflammatory, cardio-metabolic, and some malignant diseases (Table 1). CCL2 is implicated in pathogeneses of several diseases characterized by monocytic infiltrates, such as psoriasis, rheumatoid arthritis, and atherosclerosis. CCL2 is further involved in neuroinflammatory processes that mediate diseases of the central nervous system (CNS), which are characterized by neuronal degeneration. Indeed, CCL2 expression in glial cells is increased in epilepsy, brain ischemia, Alzheimer’s disease, and traumatic brain injury [161,162]. Especially, the reduction of monocyte migration to the site of inflammation by CCL2 inhibition has been shown to successfully reduce inflammation in mice [163]. In a mouse model of diabetes-associated periodontitis, oral administration of Bindarit, a CCL2 synthesis inhibitor, suppressed periodontal inflammation, resulting in reduced alveolar bone loss and increased periodontal epithelial thickness. Additionally, the monocyte infiltration into the periodontium was reduced after Bindarit treatment [163]. Patients with severe respiratory illness caused by seasonal influenza virus H7N9 show enhanced levels of pro-inflammatory factors including CCL2. In mice infected by H7N9 influenza virus, an anti-inflammatory therapy with Bindarit was demonstrated to be ineffective. Therefore, a Bindarit treatment for supportive influenza-therapy has not been pursued [164].

CCL2 may contribute to tumor progression and the spread of metastasis and could therefore be an interesting target for anti-cancer drugs. However, the CCL2/CCR2 axis seems to play a dual role in early tumor immunosurveillance and progression. Whereas the use of an anti-CCL2 monoclonal antibody reduced both the growth of primary malignant lesions and the metastases number in an implantable tumor model, CCL2 or CCR2 knockout mice developed transgenic tumors and had an increased number of metastases [165]. In multiple myeloma, CCL2 can recruit macrophages to the bone marrow. Those myeloma-associated macrophages are an essential factor in drug resistance by interacting with myeloma cells and upregulating their CCL2 expression. In turn, CCL2 upregulates the expression of MCP-1-induced protein (MCPIP1) in macrophages which triggers their polarization into the M2 phenotype that protects the myeloma cells from drug-induced apoptosis. Therefore, therapeutic strategies targeting MCPIP1 could be promising to enhance the chemotherapeutic effect [166].

Another approach to inhibit CCL2-action is to target its receptor, CCR2, to reduce inflammation. In this context, an allosteric, noncompetitive, peptidic CCR2 inhibitor, ECL1i, d(LGTFLKC) was described to solely inhibit CCL2-induced events in vitro. In a model of peritonitis, ECL1i inhibited monocyte and macrophage recruitment and further limited leukocyte recruitment and therefore disease progression in a murine model of multiple sclerosis [167]. In mice, intraperitoneal injections of TAK-779, a dual CCR2/CCR5 inhibitor, have been shown to reduce retinal vascular permeability in diabetic animals [168]. Taken together, there are several pharmacotherapeutic strategies and compounds at early phases of clinical studies (Table 1), but so far, no specific CCL2-targeting treatment has been approved for the treatment of the many diseases associated with CCL2 activation. To date, there is only one small molecule (CCR2/CCR5 antagonist Cenicriviroc) that reached a clinical phase 3 trial targeting liver fibrosis in adults with non-alcoholic fatty liver disease (NCT03028740). An antibody against CCL2 (CNTO888, Carlumab), investigated in patients with metastatic castration-resistant prostate cancer and patients with solid tumors, or an antibody against CCR2 (MLN1202, Plozalizumab), investigated in patients with RA, was not successful in clinical trials to date [169].

## 3. Summary and Conclusions

CCL2 is one of the first studied chemokines which has been predominantly investigated for the role in attracting immune cells into target tissues. The effects CCL2 are multiple and include regulation of myeloid cell function, immune response, modulation of cell-killing properties of monocytes and macrophages, but also linking obesity to its related cardio-metabolic and malignant diseases. Importantly, CCL2 exerts immunosuppressive effects that have been found to reduce the defense against malignant diseases. Targeting CCL2 signaling has attracted a lot of attention for potential clinical applications in the treatment of various types of cancer, atherosclerosis, multiple sclerosis, and type 2 diabetes. However, so far modulation of CCL2 itself or the CCL2/CCR2 axis has not yet resulted in pharmacotherapies. Several clinical trials (www.clinicaltrials.gov) with anti-CCL2 antibodies or small molecule CCR2 receptor antagonists are running and have to prove whether CCL2 is indeed a future drug target.

## Figures and Tables

**Figure 1 ijms-22-01500-f001:**
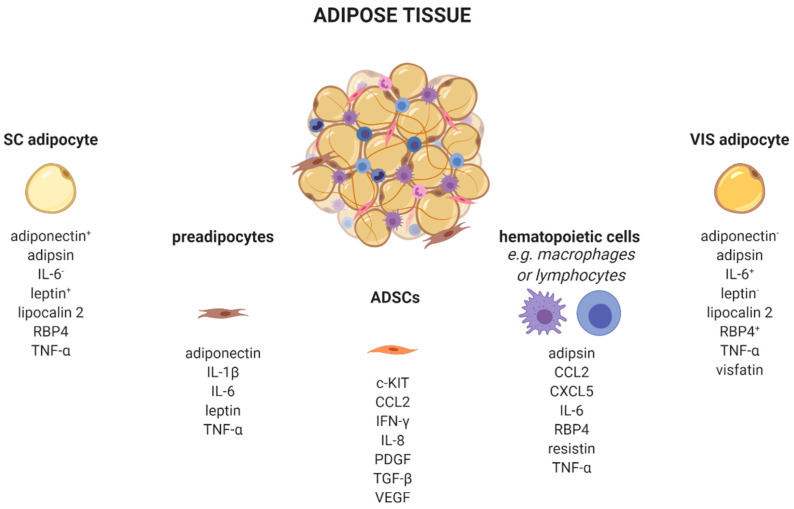
Adipose tissue cells secrete distinct adipokines. Adipose tissue consists of a variety of cell types, such as adipocytes, preadipocytes, adipose tissue-derived stem cells, and several immune cells which produce and secrete cell-type-specific adipokines. ^+/−^, higher/lower secreted in sc or vis. c-KIT, KIT proto-oncogene, receptor tyrosine kinase; CCL2, C-C motif chemokine ligand 2; CXCL5, C-X-C motif chemokine ligand 5; IFN-γ, interferon-γ; IL, interleukin; PDGF, platelet derived growth factor; RBP4, retinol binding protein 4; SC, subcutaneous; TGF-β, transforming growth factor β; TNF-α, tumor necrosis factor α; VEGF, vascular endothelial growth factor; VIS, visceral. Created with BioRender.com.

**Figure 2 ijms-22-01500-f002:**
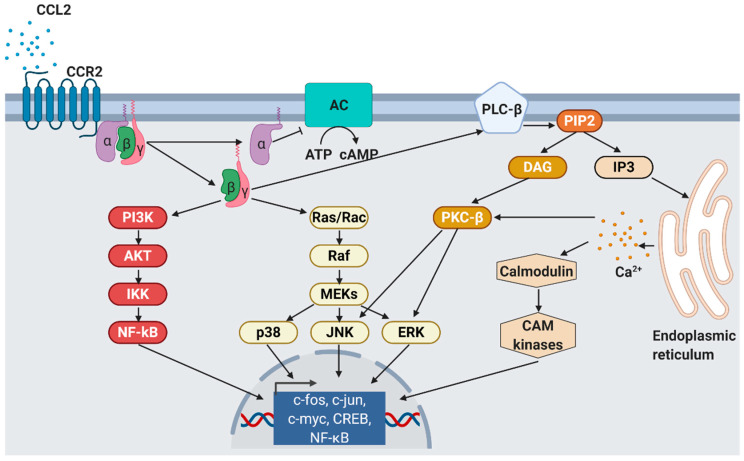
CCL2 signaling. As response to CCL2 binding at the N-terminus, extracellular loops and transmembrane bundle of CCR2, the intracellular G-protein α_i_ subunit dissociates from the CCR2 and the βγ subunit. The α subunit then inhibits adenylyl cyclase (AC) function resulting in decreased cyclic adenosine monophosphate levels. In contrast, the βγ subunit signaling induces gene expression via several pathways. PI3K, phosphoinositide 3-kinase; AKT, protein kinase B; IKK, inhibitor of NF-κB kinase; NF-κB, nuclear factor of kappa-light-chain-enhancer of activated B cells; Ras, rat sarcoma; Rac, ras-related C3 botulinum toxin substrate; Raf, rapidly accelerated fibrosarcoma; MEK, mitogen-activated protein kinase; p38, mitogen-activated protein kinase; JNK, c-jun N-terminal kinase; ERK, extracellular signal-regulated kinase; PLC-β, phospholipase C-β; PIP2, phosphatidylinositol 4,5-bisphosphate; DAG, diacylglycerol; IP3, inositol 1,4,5-triphosphate; PKC-β, protein kinase C-β; CAM, Ca^2+^/calmodulin-dependent protein kinase; c-fos, proto-oncogene c-Fos; c-jun, proto-oncogene Jun; c-myc, proto-oncogene Myc; CREB, cAMP response element-binding protein. Modified from Bose S. & Cho J. [75] using BioRender.com.

**Figure 3 ijms-22-01500-f003:**
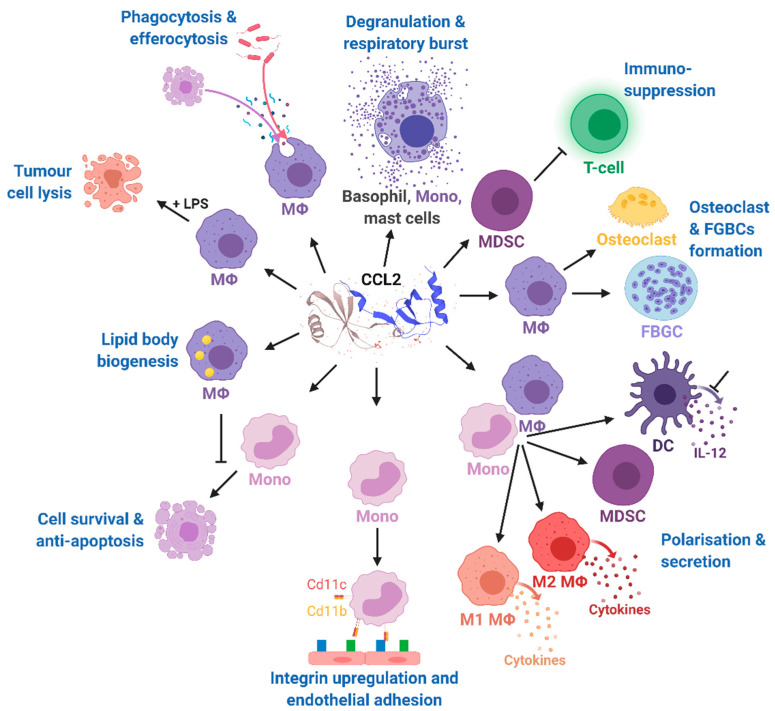
Effects of CCL2 on different immune cell types. Mono, monocytes; MΦ, macrophages; DC, dendritic cells; FBGCs, foreign body giant cells; MDSC, myeloid-derived suppressor cell; IL-12, interleukin 12; LPS, lipopolysaccharide; CD11b, cluster of differentiation molecule 11B; CD11c, cluster of differentiation molecule 11C. The CCL2 structure was taken from uniprot [86]. Modified from Gschwandtner M. et al. [60] using BioRender.com.

**Figure 4 ijms-22-01500-f004:**
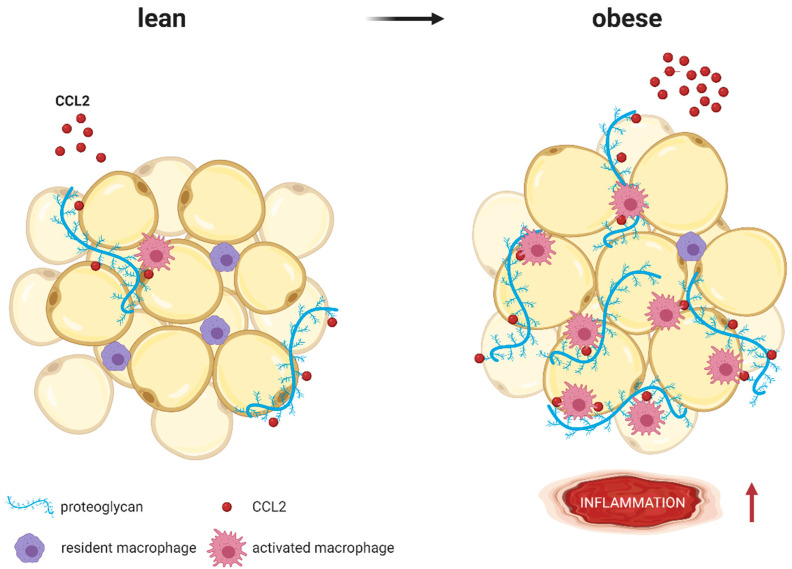
CCL2 (red molecules) binds to proteoglycans (blue strands) such as heparan sulfate or heparin, which are part of the extracellular matrix surrounding adipocytes. Whereas lean AT expresses low proteoglycan levels, expression increases with obesity. As coreceptors, proteoglycans immobilize CCL2 and present it to macrophages, resulting in higher AT inflammation (represented by **↑**). Modified from Pessentheiner A. et al. [140] using BioRender.com.

**Table 1 ijms-22-01500-t001:** Potential for CCL2 as a drug target to treat inflammatory, cardio-metabolic, and distinct malignant disease in the future. As of January 2021, no specific CCL2-targeting treatment has been approved by responsible legal authorities.

Therapeutic	Dose	Therapeutic Target/Disease	Effect	Literature
anti-human CCL2 Ab (ABN912)	0.3/1.0/3.0/10.0 mg/kg	rheumatoid arthritis	no benefits compared to placebo; dose-related CCL2 increase	[114]
anti-human CCL2 Ab (CNTO888)	2 mg/kg, twice a week	prostate cancer	47% reduced tumor burden	[112]
anti-human CCL2 Ab (MAB279)	0.3 mg/kg/day over 4 weeks	breast tumor xenograft	no effects of tumor growth or angiogenesis and on macrophage recruitment	[111]
anti-mouse CCL2 Ab	2 mg/kg/dose, twice a week	mouse and human glioma xenografts	life prolonged modestly	[110]
anti-mouse CCL2 Ab (C1142)	2 mg/kg, twice a week	hepatitis and hepatocellular carcinoma	suppressed liver inflammation and damage; reduced carcinoma incidence; reduced inflammatory markers	[113]
anti-mouse CCL2 Ab + temozolomide	2 mg/kg/dose, twice a week +800 µg in 100 µL PBS	mouse and human glioma xenografts	significantly prolonged life	[110]
Bindarit	50 mg/kg, daily	diabetes-associated periodontitis	suppressed inflammation; reduced monocyte infiltration; reduced alveolar bone loss; increased epithelial thickness	[150]
Bindarit	70 mg/kg, twice daily	H7N9 virus influenza	no anti-inflammatory effects	[151]
ECL1i, d(LGTFLKC)	90 µg, twice	peritonitis	limits monocyte and macrophage recruitment	[154]
TAK-779	30 mg/kg, daily	diabetic retinopathy	reduces retinal vascular permeability	[155]

Ab, antibody; PBS, phosphate-buffered saline.

## Data Availability

No new data were created or analyzed in this study. Data sharing is not applicable to this article.

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
