# Peer review of "Does C-C Motif Chemokine Ligand 2 (CCL2) Link Obesity to a Pro-Inflammatory State?"

_ijms, 2021, doi:10.3390/ijms22031500_

Round 1

Reviewer 1 Report

This review provides a good overview of the field of adipokines, even for the non-expert in the field, and is truly informative also for those who are experts in the field - especially the section on CCL2. The authors require approx. 3 pages as overview, before they specialize on CCL2. This appears appropriate and omits and lengthy discussions on other factors that are not in focus. 

There are also no flaws in the english language use, and which requires almost no fixing. 

Furthermore, the content is very well researched, and the many citations are a comprehensive overview of the current literature (without apparent self-citations). 10 of these references are from the year 2020; therefore the literature research is up to date.

The figures are also professionally designed, informative without being crowded, and should be quite appealing to almost any reader. There are 4 figures in total in this article, each one of them is of rather different character and composition; together they make a very lifely impression - no unncessary repetitions etc. Nevertheless, the complexity of the ligand-receptor networks (fig2) still give you a headache but this appears unavoidable. Considering this complexity, both the wording and the matching figure give a good total overview that can be untangled by the careful (and slow) reader. Which gives the review article a scholarly flavour (in a good sense). It is indeed very well structured throughout, and the more general, introductory aspects precede the specialized, and increasingly complex, and specific sections. Thats how scholarly teaching should be done (but isnt always done). 

The weakest and least informative figure is, in my opinion, Fig. 2, as it describes well-established signalling pathways that may be common place to most readers. It could therefore be considered to omit this figure, and introduce another one instead (if available) that illustrates the often highly complex ligand-receptor and cellular connections. Maybe in the style off Fig. 3, which is very appealing (and artistic) and is at the same time quite informative. 

At some places, the information becomes somewhat intense, e.g. on page 2 when differences in cytokine expression/secretion between visceral and subcutaneous fat tissue are discussed. At such occasions, it is highly beneficial that these complex patterns are explained in the figure - which helps he reader keeping the overview. At some other spots, it is unavoidable that these connections are a bit more difficult to trace, as there is no figure. It could be considered to introduce small tables to provide a similar, clear overview for some of the elements. There are no tables at all in this article, which I find somewhat surprising. 

Personally, I found section 2.4 the most interesting paragraph, with many surprising insights and - at the same time - hinting to the potential of pharmacological interference with CCL2 actions in different diseases beyond the more obvious case of obesity. This is probably even better researched than other, somewhat more well established sections. These aspects are then continued in section 2.6. Despite the similar topic, there are very few redundencies in the sections. 

Reviewer 2 Report

This review article addresses the role of the chemokine CCL2 in obesity and related diseases. In general, this should be a useful article for the obesity research community and more generally. The majority of the manuscript presents background material, describing: adipose tissue and secreted “adipokines”; various broad information about CCL2 (expression, signalling, target cells, etc.); the effects of CCL2 deletion in mice; and the association of CCL2-related genetic mutations with certain human diseases. The sections related to CCL2 in obesity start on page 8 and cover only about 2 pages (out of 11 text pages). However, I felt that the subject is covered reasonably thoroughly. Have the authors really covered ALL the available literature on the topic? If so, they should say so. If not, I would like to see the relevant sections expanded to do this (perhaps with some of the more general sections shortened).

Minor comments

Line 31: Delete the word “are”

Line 80: “induces”, not “induce”

Line 103: I suggest replacing “molecules” with “proteins”

Line 108: “overlapped” is not the right word here; perhaps “packed against”

Line 110. The gene is located on chromosome 17, not the protein (as written)

Lines 121-123: The sentence would fit better in paragraph 1 of this section.

Line 141: “becomes”, not “became”

Lines 165 and 178: “N-terminal binding”; actually CCL2 binds to the N-terminus, extracellular loops and transmembrane bundle of CCR2, not just the N-terminus.

Lines 213-216: This sentence doesn’t read quite right; what does “they” refer to in the middle?

Lines 282-3: Increased in what condition/treatment?

Lines 366-371: This paragraph requires an introductory sentence explaining the relationship of cardiovascular disease to obesity.

In several places the authors use “what”, when they should be using “which”. Some examples are on lines 135, 145, 300, 333, 410 (but there are probably others).
